

# Influence of DNA extraction kits on freshwater fungal DNA metabarcoding

Shunsuke Matsuoka[1], Yoriko Sugiyama[2], Mariko Nagano[3] and Hideyuki Doi[4]

[1] Field Science Education and Research Center, Kyoto University, Kyoto, Japan
[2] Graduate School of Human and Environmental Studies, Kyoto University, Kyoto, Japan
[3] Department of Bioenvironmental Design, Kyoto University of Advanced Science, Kameoka, Japan
[4] Graduate School of Information Science, University of Hyogo, Kobe, Japan

## ABSTRACT

**Background**. Environmental DNA (eDNA) metabarcoding is a common technique for efficient biodiversity monitoring, especially of microbes. Recently, the usefulness of aquatic eDNA in monitoring the diversity of both terrestrial and aquatic fungi has been suggested. In eDNA studies, different experimental factors, such as DNA extraction kits or methods, can affect the subsequent analyses and the results of DNA metabarcoding. However, few methodological studies have been carried out on eDNA of fungi, and little is known about how experimental procedures can affect the results of biodiversity analysis. In this study, we focused on the effect of DNA extraction method on fungal DNA metabarcoding using freshwater samples obtained from rivers and lakes.
**Methods**. DNA was extracted from freshwater samples using the DNeasy PowerSoil kit, which is mainly used to extract microbial DNA from soil, and the DNeasy Blood & Tissue kit, which is commonly used for eDNA studies on animals. We then compared PCR inhibition and fungal DNA metabarcoding results; *i.e.,* operational taxonomic unit (OTU) number and composition of the extracted samples.
**Results**. No PCR inhibition was detected in any of the samples, and no significant differences in the number of OTUs and OTU compositions were detected between the samples processed using different kits. These results indicate that both DNA extraction kits may provide similar diversity results for the river and lake samples evaluated in this study. Therefore, it may be possible to evaluate the diversity of fungi using a unified experimental method, even with samples obtained for diversity studies on other taxa such as those of animals.

# INTRODUCTION

DNA metabarcoding is a common and widely used technique in microbial diversity studies. DNA metabarcoding for environmental DNA in water (eDNA metabarcoding, (*Taberlet et al., 2012*)) has exploded in recent years as a simple and powerful method for assessing and monitoring aquatic biodiversity including microbes (*Yamamoto et al., 2017*; *Leduc et al., 2019*; *Miya, Gotoh & Sado, 2020*; *Antich et al., 2021*; *Doi et al., 2021*; *Gehri et al., 2021*). Studies on eDNA metabarcoding include methodological considerations of experimental procedures and processes, such as sampling and DNA extraction methods (*Hermans, Buckley & Lear, 2018*; *Doi et al., 2019*); *Coutant et al., 2021*; *Mathon et al., 2021*), as well as

Corresponding author
Shunsuke Matsuoka,
code_matuoka@hotmail.com

those applied to monitoring in various taxonomic groups and ecosystems (*Leduc et al., 2019*; *Ruppert, Kline & Rahman, 2019*; *Mathieu et al., 2020*; *Tsuji et al., 2019*; *Antich et al., 2021*). These experimental conditions have so far been studied mainly for aquatic animals, especially fish, and have contributed to the development of protocols for biodiversity monitoring using eDNA (*Laramie et al., 2015*; *Minamoto et al., 2021*). However, to our knowledge, there are fewer such methodological studies in fungi (*Hermans, Buckley & Lear, 2018*), and thus the potential effects of experimental procedures on the results of aquatic eDNA metabarcoding for fungi remain unclear.

Fungi drive ecosystem processes in aquatic and terrestrial habitats through the decomposition of organic matter, parasites, and symbionts of other organisms (*Peay, Kennedy & Talbot, 2016*; *Grossart et al., 2019*). However, the high local diversity and substrate heterogeneity of fungi make it difficult to assess spatial and temporal changes in their diversity, especially on large spatiotemporal scales. Recently, the analysis of fungal DNA in freshwater rivers and lakes has shown the possibility of efficiently assessing fungal diversity in the surrounding ecosystems, including aquatic and terrestrial fungi (*Deiner et al., 2016*; *Khomich et al., 2017*; *LeBrun et al., 2018*; *Matsuoka et al., 2019*; *Matsuoka et al., 2021*). Knowledge of the potential effects of experimental procedures on the metabarcoding results is essential for further validation of the usefulness of eDNA analysis in fungal diversity assessment. Among the eDNA experiments, the DNA extraction method is one of the most important factors influencing metabarcoding results (*Ushio, 2019*). For example, DNA metabarcoding results (species richness and composition detected) can vary depending on the kits or methods used for DNA extraction in experiments with terrestrial substrates (*e.g.*, mycorrhizal fungi; *Tedersoo et al., 2010* and soil; *Dopheide et al., 2019*). However, few studies have evaluated fungal DNA in freshwater (*Hermans, Buckley & Lear, 2018*).

Freshwater ecosystems to which eDNA metabarcoding has been applied are rivers, lakes, ponds, and wetlands (*e.g.*, *McKee et al., 2015*; *Deiner et al., 2016*; *Khomich et al., 2017*; *Antich et al., 2021*; *Doi et al., 2021*; *Gehri et al., 2021*). Although they differ in the presence or absence of constant water flow, there are no examples of how these habitat types affect fungal DNA extraction. PCR inhibitors could be a factor affecting DNA metabarcoding in these habitats. Previous studies on water samples from low-flow environments have shown that inhibitors carried over from DNA extraction obstruct PCR and limit the number of amplified templates that can be successfully sequenced and analyzed in downstream biodiversity analyses (*Fujii et al., 2019*; *Takasaki et al., 2021*). Examples of PCR inhibitors include humic acids derived from decomposed litters, polyphenols, and polysaccharides derived from plant and fungal cells, which suppress the activity of DNA polymerase in PCR (*Schrader et al., 2012*). The presence of such inhibitors in DNA extracts must be addressed by some means, such as dilution or purification of the template or addition of compounds (*e.g.*, dimethyl sulfoxide) that remove the inhibitors (*Schrader et al., 2012*). There are many considerations for unifying analytical methods. Inhibitors can be removed to some extent using efficient DNA extraction kits, but the ability to remove them varies between different DNA extraction kits, which may affect the results of DNA metabarcoding.

The purpose of this study was to evaluate the effect of DNA extraction kits on the results of fungal DNA metabarcoding in freshwater samples, with a particular focus on two

habitats with different characteristics: rivers and lakes/ponds. We addressed the following questions: (1) whether the degree of PCR inhibition and (2) DNA metabarcoding results (*i.e.,* species number and composition) differed depending on the DNA extraction kit used and (3) whether these effects vary by habitat type. DNA extraction was carried out on samples obtained from rivers and lakes using the DNeasy PowerSoil kit and Blood & Tissue kit (Qiagen, Hilden, Germany). The former kit has a good ability to remove inhibitors, such as humic acids, and is widely used to extract fungal DNA from soil (*Song et al., 2015*). The latter is less effective than DNeasy PowerSoil in its ability to remove inhibitors but requires fewer steps and is widely used for eDNA studies of aquatic animals (*Minamoto et al., 2021*). We hypothesized that samples extracted with the PowerSoil kit would be less affected by PCR inhibition and detect more operational taxonomic units (OTUs) than those with the Blood & Tissue kit. We also assumed that this pattern would be more evident in lakes where DNA inhibitors are considered to be more abundant than in rivers.

## MATERIALS & METHODS

### Sampling and DNA extraction

One liter of surface water was collected from each of three rivers (Kamo (Kyoto), Umekoji (Kyoto), and Toga (Hyogo)) and three lakes/ponds (Biwa (Shiga), Takaraga (Kyoto), and Shuhou (Hyogo)) in Japan from April 29 to May 18, 2017 (Fig. 1 and Table S1). Water samples were collected with DNA-free polypropylene bottles. Benzalkonium chloride (one mL, 10% w/v) was added to each liter of water sample immediately after collection to avoid a decrease in eDNA concentration (*Yamanaka et al., 2017*). Benzalkonium is thought to suppress eDNA degradation by decreasing the cellular function of bacteria and the decomposition of eDNA (*Yamanaka et al., 2017*). The water samples were kept in a cooler bag with ice packs during transport. The samples were filtered through GF/F glass filters (pore size 0.7 μm; GE Healthcare, Little Chalfont, UK) immediately after returning to the laboratory. The sample collection and filtration took approximately five minutes. The filters were stored at −20 °C until DNA extraction.

For each sample, DNA extraction was performed using three different methods (Fig. 2). First, each filter was cut into three pieces. Further, DNA extraction was performed on each piece using the following methods: (i) DNeasy Blood & Tissue kit (BT) (Qiagen, Hilden, Germany), following the same extraction protocol for MiFish analysis (*Uchii, Doi & Minamoto, 2016*). Briefly, each filter was incubated in 400 μl of Buffer AL and 40 μl of Proteinase K. Then, 220 μl of TE buffer was added to the filter and centrifuged. The dissolved DNA in the eluted solution was purified according to the manufacturer's protocol; (ii) DNeasy Blood & Tissue kit, after cutting the filter into small pieces (less than one mm) using bleached dissecting scissors (BTC); and (iii) DNeasy PowerSoil kit (PS) (Qiagen, Hilden, Germany), after cutting the filter into small pieces using bleached dissecting scissors before bead beating. Typically, in DNA extraction using BT, the filter is not cut. However, with PS, the filter was cut beforehand to allow beads to move in the bead-beating process. We performed the BTC process in step (ii) to examine whether the filter-cutting process could affect the results. DNA was extracted using each kit, according
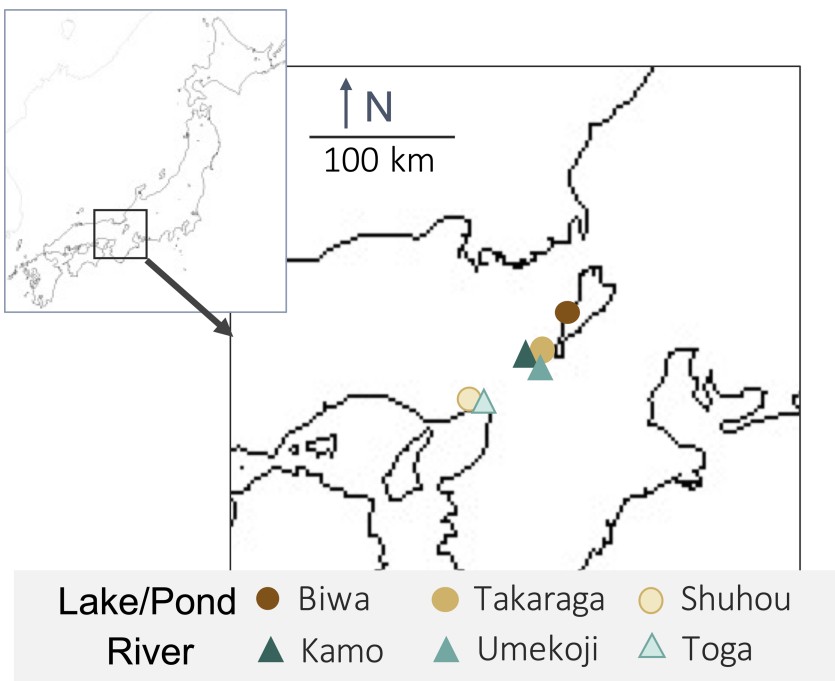

**Figure 1** **Map of sampling locations and their habitat types.** Upper-left box shows Japan and lower right box shows sampling sites.

to the manufacturer's instructions. The physical-chemical properties of the water samples and the quality of the DNA extracts (*e.g.*, DNA concentration) were not measured in this study.

## Molecular experiments and bioinformatics

Molecular biology and bioinformatics procedures were the same as those described by *Matsuoka et al. (2021)*. Briefly, the first-round PCR (first PCR) amplified the fungal internal transcribed spacer 1 (ITS1) region using the ITS1F-ITS2 primer set (*Toju et al., 2012*). An Illumina sequencing primer and six random bases (N) were combined to produce each primer. Thus, the forward primer sequence was: 5- *ACA CTC TTT CCC TAC ACG ACG CTC TTC CGA TCT* NNNNNN TAG AGG AAG TAA AAG TCG TAA -3 and the reverse primer sequence was: 5- *GTG ACT GGA GTT CAG ACG TGT GCT CTT CCG ATC T* NNNNNN TTY RCT RCG TTC TTC ATC- 3. The italic and normal letters represent MiSeq sequencing primers (TruSeq) and fungi-specific primers, respectively. The six random bases (N) were used to enhance cluster separation on the flowcells during initial base call calibrations on MiSeq. The first PCR was performed in a 12 µl volume with the buffer system of KODFX NEO (TOYOBO, Osaka, Japan), which contained 2.0 µl of template DNA, 0.2 µl of KOD FX NEO, 6.0 µl of 2× buffer, 2.4 µl of dNTP, and 0.7 µl each of the two primers (five µM). The PCR conditions were as follows; an initial incubation for 2 min at 94 °C followed by 5 cycles of 10 s at 98 °C, 30 s at 68 °C for annealing and 30 s at 68 °C, 5 cycles of 10 s at 98 °C, 30 s at 65 °C and 30 s at 68 °C; 5 cycles of 10 s at 98 °C, 30 s at 62 °C and 30 s at 68 °C; 25 cycles of 10 s at 98 °C, 30 s at 59 °C and 30 s at 68 °C, and a

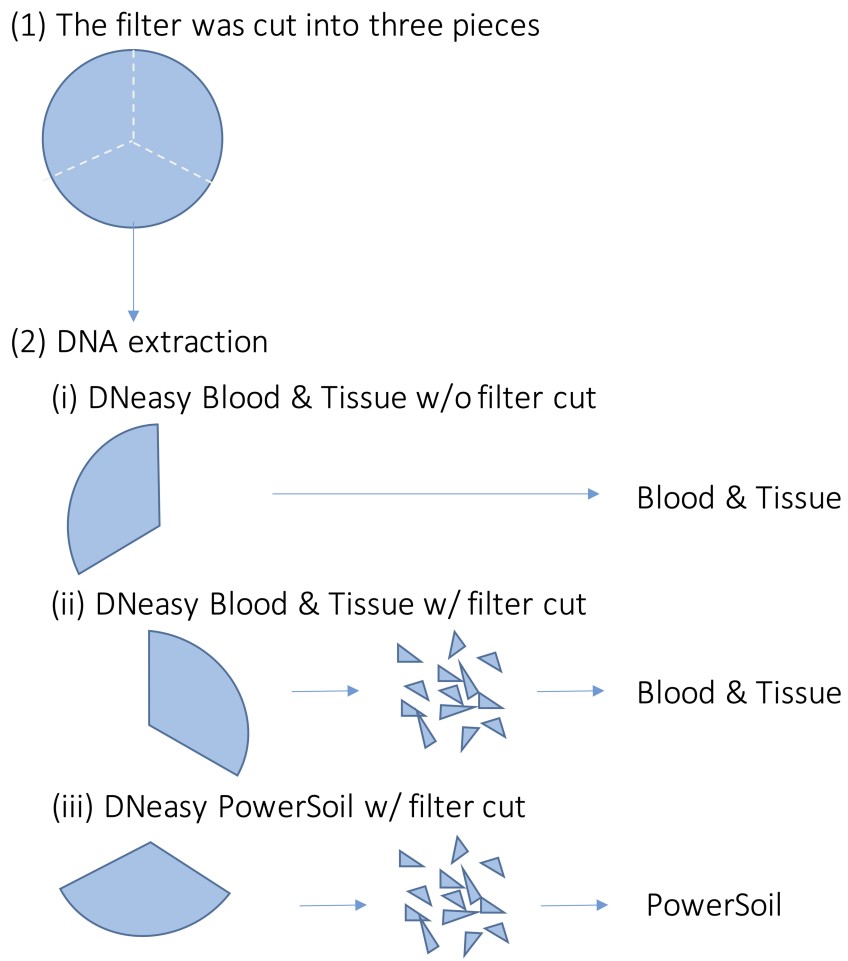

**(1) The filter was cut into three pieces**

**(2) DNA extraction**

(i) DNeasy Blood & Tissue w/o filter cut

Blood & Tissue

(ii) DNeasy Blood & Tissue w/ filter cut

Blood & Tissue

(iii) DNeasy PowerSoil w/ filter cut

PowerSoil

**Figure 2 Overview of the DNA extraction procedure.** Each filter paper was first divided into three pieces and DNA was extracted using different methods. (i) DNeasy Blood & Tissue kit without filter cut (ii) DNeasy Blood & Tissue kit with filter cut (iii) DNeasy PowerSoil kit with filter cut.

final extension of 5 min at 68 °C. Eight replicate first PCRs (per sample) were performed to mitigate the reaction-level PCR bias. Then, the replicated first PCR amplicons (per sample) were combined, resulting in a template per sample for the second PCR. The PCR templates were purified using Agencourt AMPure XP (PCR product: AMPure XP beads = 1:0.8; Beckman Coulter, Brea, California, USA) before the second PCR. A second PCR was conducted to add the index and adapter sequences. The second PCR amplified the first PCR amplicons using the primers (forward) 5-*AAT GAT ACG GCG ACC ACC GAG ATC TAC AC* XXXXXXXX TCG TCG GCA GCG TCA GAT GTG TAT AAG AGA CAG-3 and (reverse) 5-*CAA GCA GAA GAC GGC ATA CGA GAT* XXXXXXXX GTC TCG TGG GCT CGG AGA TGT GTA TAA GAG ACA G- 3. The italic and normal letters represent the MiSeqP5/P7 adapter and sequencing primers, respectively. The 8X bases represent dual-index sequences inserted to identify different samples. The second PCR was carried out with 12 µl reaction volume containing 1.0 µl of template, 6 µl of 2× KAPA HiFi

HotStart ReadyMix (KAPA Biosystems, Wilmington, Washington, USA), 1.4 µl of each primer (2.5 µM), and 2.2 µl of sterilized distilled water. The PCR conditions were as follows; an initial incubation for 3 min at 95 °C followed by 12 cycles of 20 s at 98 °C, 15 s at 72 °C for annealing and extension, and a final extension of 5 min at 72 °C. The indexed second PCR amplicons were pooled to make a library to be sequenced on MiSeq. The volume of each sample added to the library was adjusted to normalize the concentrations of each second PCR product. The pooled library was purified using Agencourt AMPure XP. A target-sized DNA of the purified library (approximately 380–510 base pairs [bp]) was then excised using E-Gel SizeSelect (ThermoFisher Scientific, Waltham, MA, USA). Sequencing was performed using MiSeq 250 paired-end sequencing at Ryukoku University, Japan. The sequence data were deposited in the Sequence Read Archive of the DNA Data Bank of Japan (Accession: DRA012030).

Low-quality sequences with a total length of less than 120 bp were excluded, forward and reverse sequences were paired, and PCR primer sequences were removed. Subsequent bioinformatics analyses were performed using commands implemented in the Claident pipeline (v0.2.2018.05.29. *Tanabe & Toju, 2013*; software available online). The removal of sequence errors was performed using the CD-HIT-OTU algorithm (*Li et al., 2012*) as implemented in claident ('clcleanseq' function with default settings). Chimeric sequence removal was performed using UCHIME v4.2.40 (*Edgar et al., 2011*) as implemented in claident ('clrunuchime' function with default settings). Operational taxonomic units (OTUs) were generated by clustering the remaining sequences based on 97% sequence similarity (*Osono, 2014*). For each OTU, the taxonomic assignment was conducted using the query-centric auto-k-nearest-neighbor method (*Tanabe & Toju, 2013*) and the lowest common ancestor algorithm (*Huson et al., 2007*) with the NCBI database. The functional guild of each fungal OTU was estimated based on the FUNGuild database (*Nguyen et al., 2016*). A matrix of OTUs × samples was then constructed (number of reads in the cells, Table S2). For the OTU matrix, cell entries with reads less than 0.0002% of the total number of reads for each sample (corresponding to two to nine reads, a criterion commonly used in eDNA metabarcoding studies, Table S2) were replaced with zero, due to such rare entries possibly representing contamination (*Lindahl et al., 2013*). Fifty-seven OTUs (22,123 reads) that were identified as non-fungal OTUs (such as zooplankton) were excluded from the analysis. The rarefaction curve showed that the number of OTUs was saturated in almost all of the samples (Fig. S1). Thus, the process of adjusting the number of reads between samples was not performed.

## PCR inhibition test

We tested the inhibition of PCR in eDNA samples. Inhibition was evaluated by including one µL of a plasmid that included an internal positive control (IPC, 207-bp; Nippon Gene Co., Ltd., Tokyo, Japan; $1.5 \times 10^2$ copies) in the PCR reaction, instead of DNA-free distilled water, and the following primer-probe set: forward primer, IPC1-5: 5-CCGAGCTTACAAGGCAGGTT-3, reverse primer, IPC1-3: 5-TGGCTCGTACACCAGCATACTAG-3′, and TaqMan probe, IPC1-Taq: 5-(FAM)-TAGCTTCAAGCATCTGGCTGTCGGC-(TAMRA)-3, which contained 900 nM of each

primer and 125 nM of TaqMan probe in a 1× PCR master mix (KOD FX Neo; TOYOBO, Osaka, Japan) and two μL of the DNA solution. The total volume of each reaction mixture was 10 μL. Real-time PCR was performed using quantitative real-time PCR (PikoReal real-time PCR; Thermo Fisher Scientific). PCR (three replicates) was performed as follows: 2 min at 50 °C, 10 min at 95 °C, and 55 cycles of 15 s at 95 °C, and 60 s at 60 °C. Non-template control (NTC) was performed in three replicates per PCR. The PCR results were analyzed using PikoReal software v. 2.2.248.601 (Thermo Fisher Scientific). The presence of PCR inhibitors was evaluated using $\Delta Ct$ ($Ct_{positive\ control} - Ct_{sample}$). $\Delta Ct$ of $\geq 3$ cycles is usually considered to be evidence of inhibition (*Hartman, Coyne & Norwood, 2005*).

## Data analysis

All analyses were performed using R v.3.4.3 (*R Core Team, 2017*). Differences in the number of OTUs between the DNA extraction kit and sampling site were evaluated. First, the effect of differences in the DNA extraction procedures was investigated using the generalized linear mixed model (GLMM) using the glmer function in the lme4 package (*Bates et al., 2015*). The error structure was a Poisson distribution, and sampling sites were specified as random terms. Next, a post-hoc test using Tukey's HSD method was performed to determine significantly different extraction procedures. The effect of sampling sites was then examined using the generalized linear model (GLM) using the glm function. The error structure is a Poisson distribution. Differences in the OTU composition by DNA extraction procedures and sampling sites were evaluated by permutational multivariate analysis of variance (PERMANOVA), using the adonis2 function in the vegan package ver. 2.5-6 (*Oksanen et al., 2020*) with 9,999 permutations. OTU compositions were evaluated using three dissimilarity indices: Bray–Curtis, Jaccard, and Raup–Crick indices. For the Bray–Curtis index, the OTU matrix containing the relative number of reads for each OTU per sample was converted into a dissimilarity matrix. For the Jaccard and Raup–Crick indices, the OTU matrix containing the presence/absence of OTUs per sample was used. The Raup-Crick index is known to be less affected by differences in the number of OTUs between samples than the Bray–Curtis and Jaccard indices (*Chase et al., 2011*). Variations in OTU composition were visualized using non–metric multidimensional scaling (NMDS) using the phyloseq package, and factors associated with the NMDS ordinations (DNA extraction procedures (PS, BT, or BTC), habitats (river or pond/lake), and latitude and longitude) were evaluated using the envfit function in the vegan package. Latitude and longitude were added to the explanatory variables in this analysis to assess whether there was any spatial structure among the samples.

Indicator taxa analysis (*De Cáceres & Legendre, 2009*) was performed to determine which OTUs had significantly different frequencies and/or occurrence between DNA extraction kits. The analysis was performed using the signassoc function in the indicspecies package ver. 1.7.8 (*De Cáceres & Legendre, 2009*) on the two OTU matrices: the relative number of reads for each OTU per sample and the presence/absence of data. We used mode = 1 (group-based) and calculated the *P*-values with 999 permutations after Sidak's correction of multiple testing.

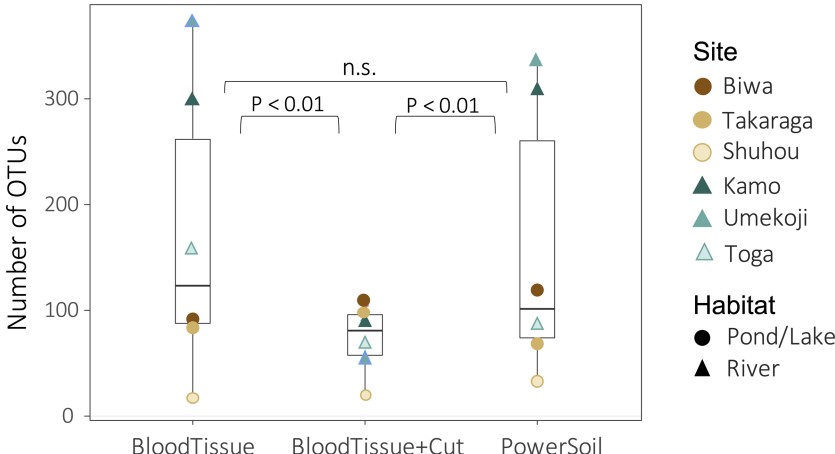

**Figure 3** **Box plots showing the number of OTUs for each DNA extraction kit.** Points indicate the number of OTUs per study site. Although no significant difference was detected between PowerSoil and Blood Tissue kits (GLMM, Tukey's HSD, $P > 0.05$), the number of operational taxonomic units (OTUs) was significantly lower in the Blood Tissue + Cut samples than in the other two treatments (GLMM, Tukey's HSD, $P < 0.01$).

## RESULTS AND DISCUSSION

A total of 1,362 OTUs were detected in 18 samples (6 sites ×3 DNA extraction conditions) (15–372 OTUs per sample; mean, 134 OTUs). Of these, 393 OTUs were assigned to Ascomycota (28.9% of the total number of fungal OTUs), 95 OTUs to Basidiomycota (7.0%), 44 OTUs to Chytridiomycota (3.2%), and 13 OTUs to Blastocladiomycota or Mucoromycota. The remaining 817 OTUs were not assigned to any phyla. FUNGuild assigned 197 OTUs (14.5% of the total number of OTUs) to functional guilds, 86 OTUs of which were saprotrophs; the others included parasites (24 OTUs) and mutualists (10 OTUs) (Table S2). The remaining 77 OTUs included endophytes and those with multiple functions. These OTUs included both those considered to be aquatic species (*e.g.*, aquatic hyphomycetes: *Tetracladium* sp., OTU_1242) and terrestrial species (*e.g.*, ectomycorrhizal fungi: *Cortinarius* sp., OTU_1248, and OTU_1346, plant pathogen: *Olpidium brassicae*, OTU_1216). These results indicate that freshwater bodies may contain DNA from both fungi living in water and those living in the surrounding terrestrial habitats (*Deiner et al., 2016*; *Khomich et al., 2017*; *Matsuoka et al., 2021*). Detailed information on the taxonomic and functional groups among the fungal OTUs is presented in Table S2 and Fig. S2.

Both DNA extraction and the study site significantly affected the number of OTUs (DNA extraction: GLMM, degree of freedom (df) = 2, deviance = 385.26, $P < 0.0001$; site: GLM, $df = 5$, deviance = 969.91, $P < 0.0001$). There was no significant difference in the number of OTUs detected between the PowerSoil kit (PS) and the Blood & Tissue kit (BT) (Fig. 3, GLMM, Tukey's HSD, $P = 0.295$), but when the Blood & Tissue kit was subjected to filter cutting (BTC), the number of OTUs detected was significantly lower than that of PS or BT (Fig. 3, GLMM, Tukey's HSD, $P < 0.0001$).

For community composition, no effect of DNA extraction condition was detected for Bray–Curtis, Jaccard, or Raup-Crick indices (PERMANOVA, $P > 0.900$, Table S3), and the variation in community composition were significantly smaller within sites than between sites (PERMANOVA, $P < 0.0001$, Table S3). The results of the NMDS ordination showed a similar trend (Fig. 4). No significant relationships with the DNA extraction method and with latitude/longitude were detected (envfit, $P > 0.100$, Table S4), and only habitat type was significantly related to the ordination (envfit, $P < 0.001$, Fig. 4, Table S4). The variation within the same study site (*i.e.,* variations among DNA extraction methods) was smaller for the Raup-Crick index than for the Bray–Curtis or Jaccard indices (Fig. 4), suggesting that the variation in OTU compositions among the extraction methods at the same study site was mainly due to differences in the number of detected OTUs rather than OTU turnover because the Raup-Crick index was less affected by alpha diversity gradients among samples than the Bray–Curtis or Jaccard indices (*Chase et al., 2011*). Furthermore, no individual OTUs were found to be significantly more abundant in a particular DNA extraction kit (indicator taxa analysis, $P > 0.05$, Table S5). These results indicate that the effect of the DNA extraction kit on the number of detected OTUs and OTU composition is limited, and that both DNA extraction kits may provide similar results that reflect differences in sites and habitats, especially in the analysis of OTU richness and composition.

PCR inhibitor tests using real-time PCR showed that the range of the average $\Delta Ct$ per site is from $-0.28$ to $0.38$ (all values are shown in Table S6), indicating that there is no inhibition in the samples ($\Delta Ct \geq 3$ is usually considered to be evidence of inhibition, (*Hartman, Coyne & Norwood, 2005*)). This result indicates that PCR inhibition may not be affected by the DNA extraction kit in the present water samples.

Differences in DNA extraction methods and kits have been shown to affect the fungal OTU composition in terrestrial substrates (*Tedersoo et al., 2010*; *Dopheide et al., 2019*). On the other hand, for fungal DNA in freshwater environments, comparable OTU composition data may be obtained between the DNA extraction kits used in this study. The PowerSoil kit has a superior ability to physically disrupt fungal cell walls and remove PCR inhibitors, although the experimental procedure is more complicated than the Blood & Tissue kit. Therefore, fungal DNA in terrestrial substrates, especially in soil, is often extracted using the PowerSoil kit. The results of the present study indicate that fungal eDNA in freshwater can be extracted using the Blood & Tissue kit, and that the same extracted sample can be used for diversity analysis of multiple taxonomic groups such as animals and fungi. These results differed from our hypothesis. However, when DNA extraction is performed with the Blood & Tissue kit, the number of OTUs detected may decrease when the filter is chopped up. The reason for the decrease in the number of OTUs detected by the filter cutting process is not clear in this study, but it is possible that the finer filter paper tends to clog the column, thereby reducing the efficiency of DNA extraction. It is better to follow the existing eDNA protocol (*Uchii, Doi & Minamoto, 2016*; *Minamoto et al., 2021*) when using the Blood & Tissue kit. In the present study, many OTUs were not assigned to taxonomic or functional groups, and the effect of the DNA extraction kit on the detection of specific taxonomic or functional groups could not be tested, which are future research topics. For example, early-diverging fungal lineages (EDF) such as Rozellomycota/Cryptomycota are

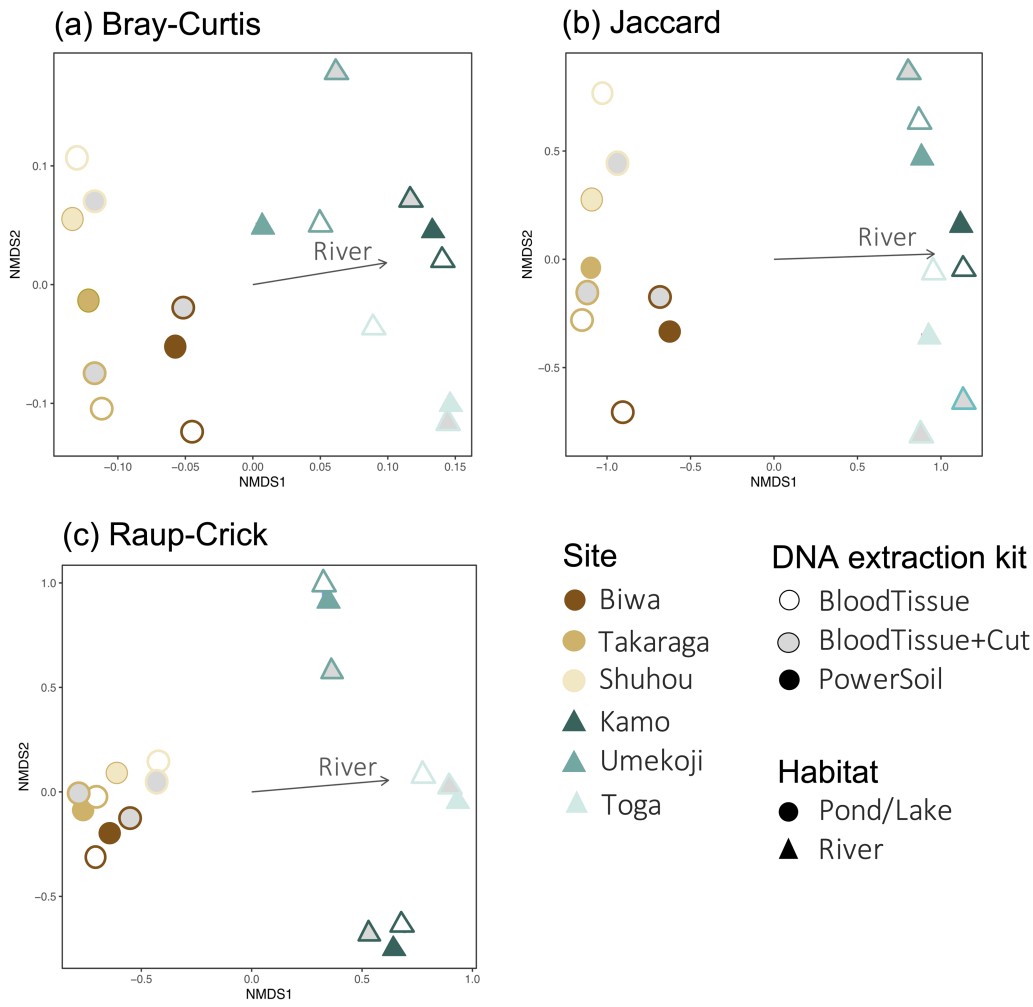

**Figure 4** **Dissimilarity in the fungal DNA assemblages among samples, as revealed *via* non metric multidimensional scaling (NMDS) ordination.** (A) Bray–Curtis index, stress = 0.1402, (B) Jaccard index, stress = 0.1048, (C) Raup-Crick index, stress = 0.1245. For all dissimilarity indices, only habitat type was significantly related to the ordination (envfit, $P < 0.05$), and DNA extraction kit and latitude/longitude were not significantly related (envfit, $P > 0.05$).

the common groups in freshwater, but information on these fungi (especially sequences of ITS regions) is more limited in the database than for the Dikarya (*Grossart et al., 2019*). Therefore, the present study may have underestimated the diversity of EDFs in particular.

It should be noted that the number of samples and sites used in this study was limited, and there is still room for consideration when interpreting the results and their generality. First, we did not replicate the data at the same site. The effect of kit differences may not have been detected because the site-to-site variation in the fungal community was greater than the variation among the DNA extraction kits. In addition, we could not statistically examine any interaction effects between the sites and the DNA extraction kits. Further validation of these findings is required by increasing the number of replicates within sites. Furthermore, the previous studies have shown that the kit with the best DNA extraction

efficiency can vary depending on the sample characteristics and the environment (*Hermans, Buckley & Lear, 2018*). Although no effect of extraction-kit differences was detected in the freshwater samples in this study, kit differences may occur in backwater lakes (*Fujii et al., 2019*), where inhibitory substances are abundant. In the future, the generality of the present results should be evaluated by analyzing the physical-chemical properties of the target water samples and assessing the quality of the DNA extracts (DNA purity and quantity). Also, comparative studies with other water filtration methods (*e.g.*, Sterivex filter cartridge; Merck Millipore) would also extend the generality.

## ACKNOWLEDGEMENTS

MiSeq sequencing was conducted in the Department of Environmental Solution Technology, Faculty of Science and Technology, Ryukoku University. We thank Hiroki Yamanaka and Hirotoshi Sato for supporting the MiSeq sequencing.

### Funding

This study received financial support from the Japan Society for the Promotion of Science (JSPS) to Shunsuke Matsuoka (20J01732) and the Environmental Research and Technology Development Fund (JPMEERF20164002). The funders had no role in study design, data collection and analysis, decision to publish, or preparation of the manuscript.

### Grant Disclosures

The following grant information was disclosed by the authors:
The Japan Society for the Promotion of Science (JSPS) to Shunsuke Matsuoka: 20J01732.
The Environmental Research and Technology Development Fund: JPMEERF20164002.

### Competing Interests

The authors declare there are no competing interests.

### Author Contributions

- Shunsuke Matsuoka conceived and designed the experiments, performed the experiments, analyzed the data, prepared figures and/or tables, authored or reviewed drafts of the article, and approved the final draft.
- Yoriko Sugiyama performed the experiments, analyzed the data, authored or reviewed drafts of the article, and approved the final draft.
- Mariko Nagano performed the experiments, authored or reviewed drafts of the article, and approved the final draft.
- Hideyuki Doi conceived and designed the experiments, authored or reviewed drafts of the article, and approved the final draft.

### DNA Deposition

The following information was supplied regarding the deposition of DNA sequences:
The sequence data is available at the Sequence Read Archive of the DNA Data Bank of Japan: DRA012030.
## Data Availability

The raw measurements are provided in the Supplementary Files.

## Supplemental Information

Supplemental information for this article can be found online at http://dx.doi.org/10.7717/peerj.13477#supplemental-information.

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
