# Peer review of "Influence of DNA extraction kits on freshwater fungal DNA metabarcoding"

_PeerJ, doi:10.7717/peerj.13477_

## Round 0.1 · original submission · Major Revisions

All reviewers agree that the paper has merit, but requires revision before it can be accepted for publication. All three reviewers have made extensive suggestions, and I would request that these be addressed in the revised manuscript.

Reviewer 1 ·

Basic reporting

The manuscript is interesting, they evaluated the effect of DNA extraction kits on the fungal DNA metabarcoding for freshwater samples. They have evaluated 3 DNA extraction methods with few sample sites in Japan. The manuscript is clear and generally well written. They mentioned that limited information are available in studies of fungi using metabarcoding in freshwater, but believe they can find more references and some externally than Japanese publications. Ex Coble et al, 2019 P1060 indicated some publications in it https://www.fs.fed.us/pnw/pubs/journals/pnw_2019_coble001.pdf . The structure of the manuscript is well done and they indicated all elements in the supplement files. I just find a bit disappointing in the analysis with all genus, species and taxonomy determination of fungal group that nothing is stated in the manuscript about organisms or group of organisms Fungal group family in the analysis. You have the info in Supple. Table S2 but nothing about it in the manuscript. The manuscript will benefits to have some abundance figure about the finding and limitation as well on the identification to the Phylum leve. genus, species....? They did great statistical analysis and well used the diversity analysis but they can I believe can go more on the analysis.

Experimental design

The disigned seem good, they have a limited number of samples and they indicated the limitation in this. However due to limited studies in the field this well presented. They present a good bioinformatics pipeline and statistics, but I think they can go more far in the analysis with the Abundance taxa figures and indication of key species. They can also add a FunGUILD analysis to their data to separate the fungal ID in different group by niches. Do you have more saprotroph, pathogen... It seem that only evaluating the overall diversity seem limited in it. Do you have any important group of fungi that could have been missed? The inclusion of Internal control with qPCR is great.

Validity of the findings

The validity of the finding seems correct. They dont have a clear answer about the cut filter why more low, could be interesting to highlight what is detected specifically there? The stats seems robust.

Additional comments

The manuscript by itself is well done, clear and they present a good statistical method to reach their conclusion. However I think more analysis in taxonomic side can be accomplish with the abundance to get a better description and evaluation of the findings. Maybe some group of organisms can show a different pattern?
Specific comments:
Abstract-Methods, first time: need a whitespace between "extractmicrobial" extract microbial
L102, should put a short description of the DNA extraction, not just reference, also when reference can be confusing to get 2 references about a protocol, Ex. L103 and 111 why 2 reference seems self citing?
L118, any URL link?
L133, why do you say no rarefaction curve were performed, the figure S1 is a rarefaction curve, not sure I follow here why not? Also seem you can indicated at what number of sequences you are getting saturated, maybe 2000, 5000 sequences?
L236, will be interested to have info on Phyllum difference before to state this sentence. Any diff in niches?

Reviewer 2 ·

Excellent Review

This review has been rated excellent by staff (in the top 15% of reviews)
EDITOR COMMENT
Thank you for making the effort to carry out a thorough review on this paper. Your review was balanced and positive, and will result in a greatly enhanced manuscript. Thank you for your considerable efforts to ensure that the paper meets the PeerJ standard.

Basic reporting

Background literature should acknowledge that the technique currently referred to as 'DNA metabarcoding' is not as new as the authors imply. In the fields of molecular ecology and microbial/fungal ecology, this was referred to as 'cultivation independent', 'marker gene sequencing', 'amplicon sequencing' approaches. The first published references to similar work occurred in the 1980's (analysis of microbes from sea water) and it became more routine in the 2000's when soil DNA extraction kits were commercialized. See additional comments for details.

Experimental design

Experimental design seems to be optimized to detect differences among sites not DNA extraction protocols (only one replicate per a site). I don't think this should prevent publication of this paper, but it should be acknowledged by the authors. The current experimental design seems underpowered to detect protocol differences. A power analysis could provide some insight. See additional comments for details.

Validity of the findings

These findings show that similar OTU richness and beta diversity patterns are detected in freshwater fungi using 2 different DNA extraction kits when following the manufacturer's protocol. The authors should be cautious about extrapolating these results more generally, as their experiment seems underpowered for detecting biodiversity differences due to DNA extraction kit (no replicates within sites). See additional comments for details.

Additional comments

PeerJ 66935

Summary

This paper describes the use of three different DNA extraction protocols using two different kits on freshwater fungal metabarcoding results. No significant differences in richness or beta diversity were found between the two kits when using the manufacturer’s protocol. The paper is straightforward and clearly written.

Major comments

The paper is very brief and there is plenty of room for the authors to include full methods to describe their molecular biology approach, move from Appendix to main text, and bioinformatic methods by including program parameters/settings. The paper would also benefit from the additional description of the fungal groups that were detected from these sites and from each kit such as stacked bar charts or heatmap--this information is already provided in the supplement but it would be nice to summarize this as a figure in the main text. Unfortunately, it seems that there is only one biological replicate per site-protocol combination. The authors could set up future work by calling for increased replication to enable statistical comparisons among protocols with sufficient power to detect differences in DNA extraction protocol (if they exist). Currently, the variation due to site differences seems to be greater than any protocol differences, making it difficult to detect variation due to protocol and this should be acknowledged.

Minor comments

Title - The word ‘DNA’ is repeated three times in the title, consider changing to something like ‘Influence of DNA extraction kits on freshwater fungal DNA metabarcoding.’ The sequencing of environmental DNA is implied, could also add to your list of keywords.

L17 - ‘rapidly expanding technique’. This technique has been around since the 80s and has been ‘routine’ since the 2000’s. Consider rephrasing to ‘environmental DNA metabarcoding is a common technique for ...’

L39-41 - Sequencing signature regions of DNA from environmental samples is not as new as is repeatedly stated in the abstract and here. Acknowledging this in your introduction is recommended. It would also be a good idea to cite the paper that first coined the catchy term ‘metabarcoding’ Taberlet et al. 2012.

L68 - ‘ there are no examples of ... ‘ consider adding ‘that we are aware of’.

L70 - ‘experimental processing after DNA extraction is inhibited by PCR’. Consider rephrasing to ‘inhibitors carried over from DNA extraction have been shown to inhibit PCR and limit the number of amplified template that can be successfully sequenced and analyzed in downstream biodiversity analyses’. Also consider listing what some common PCR inhibitors are, what they do (ex. impair Taq polymerase activity in PCR), and how common these inhibitors are in freshwater (compared to soil where this has been studied more intensively). This paragraph mentions humic acids twice, is this a common inhibitor in freshwater studies? Some common issues when extracting fungi are physically breaking down chitinous cell walls and co-extracting polysaccharides, consider mentioning these issues. Other common environmental DNA extraction issues includes DNA extraction efficiency, ug of DNA extracted and how much of it is high molecular weight (not degraded). Consider at least mentioning these other DNA extraction concerns as for some people this could affect choice of DNA extraction kit. Also consider mentioning some of the traditional ways researchers have handled PCR inhibitors, ex. dilution of DNA extract prior to PCR, addition of BSA, betaine, beta-mercaptoethanol, DMSO, gel extraction of extracted DNA, flocculation, etc. This would give more context to why you are focusing on DNA extraction ‘kits’, to avoid using potentially hazardous chemicals, as well as improving reproducibility and standardization. There is plenty of room to expand on these topics in the intro/discussion to give more context to the importance of your study.

L94 - Please add one line to explain how benzalkonium chloride reduces DNA degradation in addition to your citation.

L111 - There is plenty of space here to provide the PCR cocktail and thermal cycler settings in addition to your citations so that readers don’t have to look up another paper to understand what you have done.

L121 - forward and reverse reads were ‘paired’ not ‘concatenated’.

L122 - I was not aware that CD-HIT-OTU could address sequence errors, does it now include a denoising step or remove low quality bases/sequences? Please include the settings for that here. For each program used in your bioinformatic processing, include the parameters that were set (in addition to the papers you have cited so that readers don’t have to look up another paper to understand how data was processed in this study). Were the conserved SSU and 5.8S gene regions trimmed from the ITS1 spacer region? If it wasn’t done don’t worry about it now but do mention in the methods as this will affect how your OTUs clustered (not as sensitive, could probably detect slightly more OTUs if the conserved ends were removed).

Q - Did you quantify how much DNA that was extracted from each DNA extraction protocol? Please provide this info in the main text, it would give readers an idea of the type of yield they could expect from these protocols/kits. Did you take any gel pictures to see if it was high molecular weight (high quality) DNA? Did you notice any significant autofluorescence? DNA shearing? Did you attempt to assess DNA purity using any other complementary methods, ex. nanodrop to check for any obvious co-extracted proteins/carbohydrates, etc. If so, please mention in the text and consider including pictures in the supplement.

L156 - be sure to add a citation for each major R package that you used, ex. lme4

L167 - 169 - I’m not clear about the benefit of including the Raup-Crick index. Can you include another line to the text explaining why not being sensitive to differences in number of OTUs is a benefit for readers like myself who are not familiar with this index?

L183 - 15-372 OTUs per sample is a huge amount of variation that looks like it’s related to site differences as opposed to protocol differences. It would probably be a good idea to run a power analysis, I suspect this experimental design is under-powered to detect protocol differences (it looks fine to detect among-site differences).

L191 - consider changing OTU composition to ‘community composition’ here and on the next line.

L198 - because you don’t detect any significant differences among the DNA extraction protocols, consider adding a power analysis to see if you have enough data to detect a protocol difference if there was one. It could be that with the samples you have, you would only have the power to detect a very large effect. The current experimental design seems geared towards detecting among-site differences. If you include a power analysis, you could then set up your next paper by calling for the need to include additional replicates (more replicates from one site for each kit).

L215 - blood and tissue kits are not commonly used with fungi as fungi tend to have plant-like DNA extraction problems (thick cell walls, polysaccharides) whereas extracting DNA from blood is easy which probably explains why this kit is not often used on fungi. Not sure that previous use of blood and tissue kits to detect animals is relevant here.

-Please provide any additional information on data accessibility - you have already provided a DDJB accession but I was unable to find the record, can you double check that the data is public and accessible? Consider depositing your OTU sequences and OTU x sample table and R code to an online repository (Dryad, GitHub, etc.)

-This paper would benefit from further description of the fungal taxa that were detected by each kit and how they compare. This is present in the supplementary material, but the results could be summarized in a figure to present in the main text so that readers can which fungal groups were detected from water overall and by each kit, ex. heat map, network, or stacked bar plots.

Reviewer 3 ·

Basic reporting

The objective of the work developed in this manuscript was to evaluate the effect of DNA extraction kits on the results of fungal DNA metabarcoding for freshwater samples, namely of rivers and lakes/ponds. Overall, the manuscript is well written.

The introduction section cites mostly works using eDNA to target fish (e.g., L41). Although these examples are valid, I suggest author also include examples of other works focusing on other organisms like metazoan, eukaryotic communities (e.g., Leduc et al 2019; Antich et al 2021). These are just 2 examples; author may find other articles more suitable.
Also, reference support should be given at L48-50, when authors state that “there are fewer such methodological studies in fungi”; and L129-130 “which corresponds to 2–9 reads, a typically used criterion in eDNA metabarcoding studies”.

L64-65. This sentence should be rephrased. Please see paper by Hermans et al 2018, where they tested DNA extraction methods to simultaneously extract bacterial, fungal, plant, animal and fish DNA from soil, leaf litter, stream water, stream sediment, stream biofilm and kick-net samples, as well as from mock communities. This paper might be useful also to the discussion section, to compare with results obtained here.

References
Leduc, N, Lacoursière-Roussel, A, Howland, KL, et al. Comparing eDNA metabarcoding and species collection for documenting Arctic metazoan biodiversity. Environmental DNA. 2019; 1: 342– 358. https://doi.org/10.1002/edn3.35
Antich, A, Palacín, C, Cebrian, E, Golo, R, Wangensteen, OS, Turon, X. Marine biomonitoring with eDNA: Can metabarcoding of water samples cut it as a tool for surveying benthic communities?. Mol Ecol 2021; 30: 3175– 3188. https://doi.org/10.1111/mec.15641
Hermans, SM, Buckley, HL, Lear, G. Optimal extraction methods for the simultaneous analysis of DNA from diverse organisms and sample types. Mol Ecol Resour. 2018; 18: 557– 569. https://doi.org/10.1111/1755-0998.12762

Experimental design

I think it is important that authors clearly state their hypotheses in the last paragraph of the introduction. Did you expect one of the kits to perform better? Or you expected both to perform equally? Did you expect differences between rivers and lakes/ponds?

Regarding the experimental design, the most important question I see is why you tested for filter cutting or not, using the Blood and Tissues kit, but did not do the same test for the Powersoil? It is really important for the readers to understand your reasoning. Would authors also expect differences in performance?

Regarding statistics, author presented only the main effects. It could be interesting to look at the interactive effects “dna extraction x sampling site” for both number of OTUs and community composition. Also, why authors decided to test for latitude and longitude (L172) effect on OTU composition? This is not clear in the text and was not stated and an objective of the work previously. Finally, on L174. authors explain the indicator taxa analysis and they state that it “was performed to determine which OTUs had significantly different frequencies between DNA extraction kits”; but below you say that you used the two matrices, i) relative number of reads and ii) presence/absence. This seems a bit confusing since presence/absence matrix will not provide information of OUT frequencies. Please clarify this question.

Others small details below to increase clarity of methods:
L92. Do authors measured any physical-chemical parameters of the water (Table S1)? If not from this study, do author have these data from previous studies? Having information of content of organic matter for instances, would be important to support your results and discussion.
L95. How long between samples collection and filtration? Please add this information.
L96-97. Why authors selected the GF/F filter (0.7um pore size) and not Sterivex type filter (0.2um pore size)? Would you expect differences in the performance? See paper Spens et al 2017. I think it is important to explain this choice and maybe add some points to discussion.
L104. Can author give an estimate of the size of filter pieces after cut? This can be important for other researcher to replicate your study.
L127. Why did authors used the NCBI database and not other options like UNITE? Maybe add a small sentence to justify your choice.
L149. Please correct: ΔCt of ≥3 cycles and is usually considered > ΔCt of ≥3 cycles is usually considered.

References
Spens, J., Evans, A.R., Halfmaerten, D., Knudsen, S.W., Sengupta, M.E., Mak, S.S.T., Sigsgaard, E.E. and Hellström, M. (2017), Comparison of capture and storage methods for aqueous macrobial eDNA using an optimized extraction protocol: advantage of enclosed filter. Methods Ecol Evol, 8: 635-645. https://doi.org/10.1111/2041-210X.12683

Validity of the findings

I think the section results and discussion can be improved to increase the value of the manuscript. Once author include their hypotheses in the introduction, it will make the discussion clearer already.

L197-202 Authors state that “variation within the same study site (…) was smaller”. But, it is interesting to notice that overall variation among pond/lake sites was also much smaller in the Raup-Crick; but rivers did not follow that pattern. Do authors have and explanation for that?

I think the paragraphs L208 and L224 should be rephrased and discussion better integrated. With the current structure, and by the end of paragraph L208-223, it seems like Blood and Tissue kit is the best option. Just in the paragraph L224-233 author refer that water samples with high content of organic matter (for instance) might result in PCR inhibition, and so using a kit that removes more inhibitors, like the Powersoil, could be a better option. I understand the separation to show first the results for OTUs and after for PCR inhibition, but maybe integrating and discussing together can improve clarity. And, adding information on the physical-chemical characteristics of your sampling sites would definitely give support to your results.

Additional comments

Minor typos on Reference list:
L295. Systematicreview > Systematic review
L347. Ofnon-native > of non-native
L356. Year here is 2017, but on L.95 is 2016. Please confirm and correct.

---

## Round 0.2 · Minor Revisions

The reviewers have noted the significant effort you have invested to improve your paper, and have suggested further edits, focusing on minor corrections for the most part, but there are areas of methods description which require more detail, and which have implications for the results section also.

Reviewer 2 ·

Basic reporting

No comment

Experimental design

No comment

Validity of the findings

No comment

Additional comments

Peerj-66935

Summary

This paper is greatly improved over the original submission. I can see now that the study was performed in a much more stringent manner than I had originally thought. The newly added background in the introduction and details added to the molecular biology and bioinformatic steps greatly improves the paper.

Major comments

None

Minor comments

L24 - change to ....of DNA extraction... remove the word ‘the’

L29 - remove square brackets, just do ... DNA metabarcoding results; i.e., operational ...

L125 - change physicochemical to physical-chemical

L129 - change ‘Molecular experiments’ to ‘Molecular biology’

L148 - please clarify, was the second PCR to add Illumina sample indexes? If so, indicate the index kit that you used, ex. Nextera?

L286 - just a comment, there are numerous early diverging fungal lineages that have not been identified to the species level that are commonly found from freshwater samples, search Rozellomycota/Cryptomycota but all early diverging fungal lineages are difficult to classify as so many are currently unnamed (Blastocladiomycota, Chytridiomycota and everything in between). It would be interesting to know if you detected any of this highly diverse group (Cryptomycota) from your own samples as this could explain why you were unable to identify so many OTUs. In GenBank they are often only labelled as ‘uncultured fungal clone’ or ‘environmental sample’. During LCA processing, these types of vague identifications really hinder taxonomic assignment. They are best identified using phylogenetic methods with reference sequences used to represent this group (try BLASTing a few of these by hand and looking at the distance tree results provided by the NCBI BLAST method). It would actually be nice to add a few sentences about this common freshwater group to your discussion to highlight how common and diverse they are and how challenging they can be to properly identify because none of these samples actually have species names associated with them (except for the well-known chytrid parasite Rozella allomycis).

L286 - also, it would be good to add one sentence (perhaps in the methods section) to address how specific the primers are that you used in this study, are they expected to amplify non-fungal taxa such as macroinvertebrates?

Fig 1-legend. Add “Upper-left box shows Japan and lower right box shows sampling sites.” or something similar.

Reviewer 3 ·

Basic reporting

I want to acknowledge the very extensive work authors have performed in the MS. I am glad that all my questions and suggestions (and other reviewers) have been addressed. I think the MS has improved in clarity and general interest to the readers.
Authors have now included in the introduction section the questions addressed in this MS. This really helps the readers to understand the objectives. However, I think adding hypotheses (if possible) would benefit the MS. For instance, you state that you will address the following questions: “(1) whether the degree of PCR inhibition and (2) DNA metabarcoding results (i.e., species number and composition) differed depending on the DNA extraction kit used and (3) whether these effects vary by habitat type”. I understand that authors have not measured physical-chemical parameters of the water and so this data is not available. But if we think these are environmental samples (and unless these are super oligotrophic streams and lakes), we would expect some organic matter and even humic acids to be present, for instances. As so, I would hypothesize that the kit Powersoil would be more efficient preventing PCR inhibition compared to the blood and tissue. The same exercise can be made for the sites: based on your knowledge on the streams and lakes/ponds, can you hypothesize that one of these could have more inhibitors and so in one of these type of sites one of the kit would provide better results? Or characteristics are similar and you expected them to perform the same?

Experimental design

The section on methods is much improved now. One thing I am missing is description in this section of how you performed the analysis of functional traits with FUNGuild. Another thing that is missing is the description of effect of the Latitude and longitude. You say in the methods you will test this, but nothing in mentioned in the Results and discussion section. Only in the legend of Fig.4 some information was added. Please also describe it in the results section.

Validity of the findings

No comment

Additional comments

Some minor comments/suggestions:
L42. Please consider changing “as well as” to “including”.
L68. Please remove repeat word “fungi”.
L71. eDNA metabarcoding has been also applied in wetlands (e.g., https://doi.org/10.3996/042014-JFWM-034). Please provide references for other examples stated.
L81. Please provide a reference.
L122. Please correct “beforehand to allow”.
139. Consider changing “duplicated” to “replicated”
L252. If the community composition is similar, shouldn’t the P-value be >0.05? in the text you have P<0.0001. Please clarify.
L470. Please add to the legend the functional assignment of species.

---

## Round 0.3 · accepted · Accept

Congratulations on your excellent contribution.